# "Mobility as a Service" Platforms: A Critical Path towards Increasing the Sustainability of Transportation Systems

**Carlos Oliveira Cruz** [1,*]  **and Joaquim Miranda Sarmento** [2]

1    CERIS, Instituto Superior Técnico, Universidade de Lisboa, 1049-001 Lisbon, Portugal
2    ISEG-Lisbon School of Economics and Management, Universidade de Lisboa, 1200-781 Lisbon, Portugal; jsarmento@iseg.ulisboa.pt
*    Correspondence: oliveira.cruz@tecnico.ulisboa.pt

**Abstract:** Urban mobility is experiencing a profound change. Mobility patterns are becoming more complex, and typical home–work–home travel is no longer the rule, as journeys tend to connect multiple points in a rather inconstant pattern. This has changed the approach to transport planning. Existing transportation planning and operation approaches have been focussed on the ability to identify and forecast typical home–work/school–home travel and subsequently plan the transport system accordingly. The traditional approach has been: Forecast - > plan - > deliver. New mobility patterns and mobility solutions are characterised by greater flexibility, taking advantage of the "sharing concept" and simultaneously providing solutions that have lower greenhouse gas (GHG) emissions. These dynamics and an evolving environment raise several new challenges at different levels, fostering the development of Mobility-as-a-Service (MaaS). This system transforms the physical transportation system into a commodity and takes advantage of the internet of things (IoT). However, the onset of MaaS solutions is anything but linear. Several business models have emerged, with different partners originating from different industries (e.g., technological, transport operators, infrastructure managers, etc.) developing their own solutions, often in competition with others. It is not unusual to find different MaaS solutions in the same city, which integrate different solutions. This paper intends to provide an analysis on the main challenges affecting mobility in general, and MaaS in particular, as well as the main business models used for delivering MaaS solutions. The paper uses a case study in Lisbon to illustrate some of the challenges.

**Keywords:** MaaS; urban mobility; digitalisation

## 1. Introduction

Over recent years, urban mobility has experienced a profound change. Not only is it becoming more and more complex, but typically home–work–home travel is no longer the rule. Nowadays, journeys typically connect multiple points in a rather inconstant pattern [1]. This has changed the transport planning and operation approaches, which have been focussed on the ability to identify typical "home–work/school–home" travel and plan the transport system accordingly. The traditional approach has been to 1—forecast, 2—plan, and 3—deliver. However, the traditional transport system is also experiencing great change. Despite being supported by public transport (bus + metro + rail) and the private car, new mobility solutions are emerging. These are characterised by greater flexibility, in that they take advantage of the "sharing concept" (e.g., bicycles, electric scooters, car-sharing, etc.), and simultaneously providing solutions with lower greenhouse gas (GHG) emissions [2]. As a result of these developments, the traditional transport system itself is changing. As a consequence, the typical forecast–plan–deliver paradigm is progressively being abandoned.

These transformations have made urban mobility become less predictable, whereby it follows a fuzzier pattern, with urban mobility acting as an "active organism", changing and adapting to new circumstances and patterns. This is partly due to the fact that the solutions being offered to consumers are being replaced rapidly, with new options becoming obsolete over much shorter periods than previously. Furthermore, when solutions are understood to be problematic or inefficient, they are easily abandoned. For example, a few years ago, segways seemed to be the answer for short-distance urban travels, but these have now been quickly replaced by electric bicycles and/or electric scooters, at a much lower cost and higher convenience to users [3].

In terms of urban mobility, all these changes in transportation planning and transport systems, together with the inherit complexity of all the system, has raised several new challenges at different levels. For example, with regards to travel payment, the typical model of a monthly pass vs. a one-travel ticket can no longer meet the demand of less stable patterns with regards to transport utilisation, as the payment system needs to integrate different modes and mobility solutions. There is a growing need for an integrated system that enables the use of different modes, without the need for different physical tickets. However, dynamic information systems are also required—which facilitate the sharing of revenue between the distinct modes and operators.

Part of the response to the challenges posed by these new realities is the development of Mobility-as-a-Service (MaaS) solutions which: "Combine transport services from public and private transport providers through a unified gateway that creates and manages the trip, which users can pay for with a single account. Users can pay per trip or by a monthly fee for a limited distance. The key concept behind MaaS is to offer travellers mobility solutions based on their travel needs" [4].

As discussed by [5], MaaS "aims to integrate various forms of urban transportation into a single mobility service accessible on demand, which is possible due to the digitalisation of urban mobility". MaaS transforms the physical transportation system into a commodity, and thus takes advantage of the internet of things (IoT), i.e., it communicates real-time information regarding the transportation system capacity and its operation. However, the onset of Maas is anything but linear. Several business models have emerged, with different partners originating from different industries (e.g., technological, transport operators, infrastructure managers, etc.) developing their own solutions, often in competition with others [6]. It is not unusual to find different apps and platform solutions in the same city, which integrate different solutions, creating a relatively complex system for users. As discussed by [7–9], MaaS solutions are an integrator in a functional ecosystem which needs to be in place and well-connected, comprised of four major stakeholders: Customers, transport operators, a data aggregator, and a trusted MaaS advisor which manages the whole operation.

Besides facilitating payment, by eliminating physical tickets and enabling a genuinely digital utilisation system, MaaS also produces massive quantities of data, which are crucial for enabling city and transport planners to understand the dynamics of the mobility system and identify the bottlenecks of the system and act accordingly [10,11]. From a public policy perspective, this data represents the fundamental basis for informed decision making for city management. However, considering that these solutions are primarily developed by private companies, concerns are starting to be raised not just over data privacy, but also regarding data property.

In this paper, we provide a conceptual framework of the existing drivers of change behind the adoption of MaaS solutions, and also the main barriers, together with the main advantages and disadvantages of these solutions. Additionally, we provide an overview of some of the operational solutions that are already in operation.

We found that two fundamental drivers are the main reason why MaaS is so useful for consumers. Firstly, MaaS allows for a more flexible and tailor-made planning solution with an associated "mobility bill", and secondly, it provides a "one stop shop" for all mobility services.

This paper is organized as follows: After this introduction, Section 2 presents a literature review of the concept of MaaS. Section 3 analyses the main drivers of change, and Section 4 presents an

international overview of MaaS solutions. Section 5 presents the case study (Lisbon, Portugal), and, finally, Section 6 presents the main conclusions.

## 2. Concept of MaaS: A Literature Review

Nowadays, MaaS integrates any discussion, analysis, or forecast on future mobility systems. As discussed by [12] the mobility ecosystem is becoming more fragmented, with the introduction of transport solutions, new business models, and new companies—all of which are aligned with the same common objective, namely to provide affordable, convenient, and sustainable mobility solutions. This fragmentation increases proportionally to dynamics of the business environment. This means that new services often have a more complex and dynamic pricing strategy, with prices varying according to the utilization patterns and the level of usage, etc. Frequently, a more dynamic business environment also means that these new transport operators tend to be start-ups which are testing business models [13]. Some survive and grow, whilst others are merged, or fail to succeed.

MaaS provides a layer that allows for the functional integration of tariffs of the system. In theory, this one-stop-shop platform should enable the integration of all the potential different links of a single journey with different operators, and with different fare systems.

Given the extraordinary advantages of MaaS solutions, and the leverage potential for accelerating the evolution of the urban mobility ecosystem, there has been an increase in the quantity of literature on the topic, and there has also been a certain effort to conceptualise MaaS and structure the various levels of development.

Reference [14] provided a theoretical conceptualisation of the MaaS concept, arguing that it has the attributes of a "hyped' socio-technical phenomenon: "It seems to be a loosely connected patchwork of optimistic political dogma, activists' enthusiasm, anecdotal evidence of successful services and a firm belief of investors in companies such as Uber".

However, some authors adopt a more pragmatic approach to MaaS in an attempt to understand the potential business models behind these solutions, e.g., [15]. The core definition of MaaS is still not unanimous. While some authors adopt a more technologically-based definition, where MaaS is a technical system (a digital platform) [16] which enables the integration of the purchase process of several distinct journeys and/or travel elements, others use MaaS as a more broad definition for not just the technological solution, but also the entire process of transport integration and the commoditisation of transport infrastructure. By transport infrastructure, we do not simply refer to roads or railways, but also to all the "hardware" that enables the delivery of transport services, including, but not exclusive to: Vehicles, stops, stations, charging points, etc. Table 1 summarises the main definitions of MaaS found in the literature.

## 3. Drivers of Change

There is a growing number of drivers of change which affect urban mobility, and, particularly, the adoption and configuration of MaaS solutions. This section presents a brief overview of these drivers. To structure the analysis, we have classified the drivers into three categories: Technological, societal, and institutional. This analysis does not intend to provide a comprehensive and exhaustive list of all the drivers, but rather focusses on those that have the greatest potential impact. This classification is not unique, and alternative classifications can be found in the literature.

**Table 1.** Main definitions provided in the literature for Mobility-as-a-Service (Maas).

| Author | MaaS Definition |
| --- | --- |
| [17] | A distribution model that deliver users' transport needs through a single interface of a service provider. |
| [18] | MaaS is the integration of various forms of transport services into a single mobility service, which is accessible on demand. To meet a customer's request, a MaaS operator facilitates a diverse menu of transport options, be they public transport, ride-, car-, or bike-sharing, taxi or car rental/lease, or a combination thereof. |
| [17] | "Mobility as a Service (MaaS) is a mobility distribution model where a customer's major transportation needs are met through the use of one interface and are offered by a service provider. Typically, services are bundled into a package—similar to mobile phone price-plan packages." |
| [19] | A concept which allows households to purchase packages of mobility that provide an alternative to car ownership. |
| [20] | In the first place, MaaS is a distribution model for transport services. MaaS integrates transport modes through the internet. |
| [14] | MaaS is widely regarded as being the next paradigm change in transportation. Service providers are expected to offer travellers easy, flexible, reliable, price-worthy, and sustainable everyday travel, including, for example, public transport, car-sharing, car leasing, and road use, as well as more efficient options for goods shipping and delivery. |
| [21] | The concept of MaaS is relatively simple: The bundling of different transport means, public and private, into one easy-to-use package for the customer. The service is provided to the customer via mobile applications and payment is handled via a digital wallet. The actual business cases and large scale pilots of MaaS, in addition to other empirical evidences, are yet to be seen. |
| [22] | MaaS comprises a sophisticated conglomerate of heterogeneous transportation means, physical infrastructures, and information and communications technologies (ICTs) which work in combination to enable citizens to reach their destinations efficiently. |
| [23] | MaaS relies on a digital platform that integrates end-to-end trip planning, booking, electronic ticketing, and payment services across all modes of transportation, public or private. It's a marked departure from where most cities are today, and from how mobility has been delivered until now. |
| [24] | Can be thought of as a concept (a new idea for conceiving mobility), a phenomenon (occurring with the emergence of new behaviours and technologies), or as a new transport solution (which merges the different available transport modes and mobility services). |
| [15] | MaaS aims to bridge the gap between public and private transport operators at a city, intercity, and national level, through the integration of the currently fragmented tools and services required by a traveller for a trip (planning, booking, real time information, payment, and ticketing). Mobility is a user-centric, intelligent mobility distribution mode, where all mobility service providers' offerings are aggregated by a sole mobility provider—the MaaS provider, and are supplied to users through a single digital platform. |
| [25] | "Mobility as a Service (MaaS) is the seamless, infinitely adaptable delivery of mobility, together with associated travel information, necessary ticketing and payment services, across all modes of transport." |
| [26] | MaaS is a term used to describe digital services, often smartphone apps, which people use to access a range of public, shared, and private transport, using a system that integrates the planning, booking and paying for travel. |
| [27] | To meet this challenge, "Mobility as a Service" (MaaS) concepts are introduced in the market which offer an individualised one-stop access to several bundled travel services, based on customers' needs. |
| [28] | Mobility as a Service (MaaS), which uses a digital platform to bring all modes of travel together into a single on-demand service, which has received great attention and research interest. |

Reference [25] argues that the emergence of Maas reflects several distinct trends, which the author classifies into three different types. First, markets trends—as consumers are changing their patterns and preferences towards a more flexible lifestyle, and are searching for services that provide a simplified and more effective user experience. Second, the private industry vision—as within the private sector

several technological developments have occurred which facilitate the emergence of Maas, such as IoT, big data, blockchain, shared services, etc. Third, the government vision—of governments and decision-makers who develop a more integrated vision of the transportation system, where congestion management and public spending value maximisation are both important goals, as is promoting a more equitable and inclusive mobility system.

Certain technological drivers can have a substantial impact, examples being: Electrification, autonomous vehicles, shared mobility, the internet of things, blockchain, and artificial intelligence.

With regards to electrification, urban mobility is increasingly evolving towards electric systems. For instance, metro systems are growing [29], bus operators are replacing existing combustion-engine fleets with electric fleets [30], and private cars are also moving towards electric/hybrid solutions (some cities have already established areas which are exclusively for electric vehicles).

Autonomous vehicles and autonomous driving are probably the largest disruption for the automobile industry. The potential impact of such technologies ranges from vehicle and road safety to congestion mitigation [31]. Autonomous vehicles have the potential to decrease the cost of incorporating car solutions within MaaS systems. Nowadays, the use of a private car as a complement within MaaS is achieved through car-sharing (e.g., the case of DriveNow in Lisbon). Autonomous vehicles will be able to increase the efficiency of vehicle utilisation, and therefore decrease their unit costs (per km), and consequently encourage MaaS solutions (as discussed by [28]). This will lead to a strong prevalence of shared mobility.

Shared mobility will bring about a dramatic improvement in the efficiency of the utilisation of private cars (as cars are parked 95% of the time, without being used). Furthermore, shared mobility, particularly of private cars, has led to the creation of several new services and business models, examples being Uber and Cabify, and also of pure car-sharing models, such as DriveNow. However, this shared mobility does not just affect private cars, but also motorbikes (e.g., eCooltra) and electric scooters (e.g., Lime). Considering that shared mobility improves the options for mobility services and thus increases the flexibility of travel options, it provides more leverage for the adoption of MaaS solutions—leading to the establishing of more services, with distinct integrated operating logics and added value for users. New technologies will accentuate this impact, examples being: The internet of things, blockchain, and artificial intelligence. For more on shared mobility, see [32,33].

The internet of things, although much less visible to the user than, for example, shared mobility solutions, provides the backbone for the growth of MaaS solutions. One of the key principles for MaaS is to integrate the various mobility systems and solutions, not just functionally, but also in terms of fares and utilisation rules. This integration is dependent on the ability to achieve a true digital connection in order to ensure robustness and real-time tracking of transactions and their corresponding validation (blockchain technology also provides an important contribution which is described below). Furthermore, integration enables data aggregation and the development of solutions for real time travel and planning within the mobility system.

Big data and data analytics also play an important role. By aggregating data, MaaS systems provide a valuable set of data, which can be used to extract meaningful information on passengers travelling patterns, their preferences, and choices, helping decision makers to plan for better systems, but also allowing for live-feed information flows that can help the transportation system to adapt, in real time, to changing demand.

With regards to blockchain, as discussed above, in practice, a growing number of stakeholders of the new mobility ecosystem both compete and cooperate. For instance, there are private, public, or even public/private actors who generate many more transactions that the traditional system—which is based solely on public transport and car utilisation. Recording and validating these transactions is critical for ensuring the reliability of the system. Indeed, it is also important to carry this out with reasonable transaction costs. The number of transactions is increasing and will continue to increase with the diversification of mobility solutions and business models, as well as the growing complexity of mobility patterns. However, the direct benefits of blockchain are not limited to transactions alone,

as blockchain also facilitates the development of smart contracts, which allow for more effective contract management and compliance, whilst also decreasing the existing complexity of monitoring traditional "paper-based" contracts. This is particularly critical in terms of the growing complexity of public–private partnerships (PPP) and concessions contractual relationships in the urban environment (see more in [34,35]. Traditional concession models are based on a pre-determined service, for which the operator collects tolls and/or is granted subsidies from the grantor, over a specific period of time. These contracts are relatively rigid, and are based on the assumption that the service (and ridership) can be forecasted within reasonable levels of accuracy. However, the reality is that mobility systems require a much more flexible approach from transportation systems and transport operators, as well as the continuous ability to adapt the services to new users' preferences, and also the capability to adjust to the emergence of competing services. These "flexible contracts", as discussed by [36], will be much more difficult to monitor, with blockchain technology providing a valuable role in overcoming this challenge.

Finally, with regards to the technological drivers, artificial intelligence is fundamental for leveraging MaaS solutions, as it provides the ability for data generators to identify patterns and to customise the solutions on offer, and to also to provide transport operators with the ability to understand users' preferences and their mobility patterns, etc.

The following societal drivers will determine most of MaaS development in the future: More dynamic sustainability, users' preferences, urban planning, congestion and network effects, together with efficiency gains.

Sustainability is an omnipresent concept when discussing urban mobility. For whatever the changes, evolutions, or adaptations of mobility systems, it is now a mandatory pre-requisite that they guarantee an improvement in the sustainability of the system. MaaS has the potential to positively contribute to this objective for several reasons. First, MaaS has the potential to internalise the externalities of several transport modes on a global scale. This can be done through smart tariffs that can penalize those solutions that are environmentally more damaging, while cross-subsidizing those with lower impacts. Second, all-in-all, MaaS will contribute to improving the efficiency of the utilisation of existing assets and mobility systems, by allowing users to choose those solutions that maximise their utility. Third, the digitalisation of payments and tickets can significantly decrease the carbon footprint of the ticketing systems.

In addition, user preferences are constantly changing. Besides the growing concern for sustainability, nowadays users are more demanding in obtaining real time data from the system, in order to minimise waiting times and overall travel times, as well as ease of use of the system. MaaS also has the potential to facilitate users' experience, as nowadays access to real time information is typically spread over several platforms. Google is probably the data aggregator which provides the majority of information on public transport systems and traffic conditions on a worldwide scale, although several local aggregators also exist in each city which provide additional layers of information (some of which are publicly financed, such as the smart city app in Izmir Turkey). Unfortunately, these platforms are typically disconnected from the established ticket purchase and pay-by-ride platforms, which consequently creates an additional complexity to the system, as shown in the case study presented below.

One of the main challenges of urban mobility and MaaS is urban planning and congestion [37]. As mentioned by [4,38], it is not clear whether MaaS will reduce or increase traffic congestion, for as cars become more efficient, this could lead to a change in operation of public transport, as is the case of UBER and other digital car-ride platforms. Network effects and efficiency gains will also help reduce congestion, by increasing the scale of the service and also by creating a much more "tailored" network to respond to the teal needs of users. This could have the effect of aligning supply with the real demand at any moment, by understanding the patterns of demand of travel in real time. Furthermore, the public sector has the ability to encourage behaviour and create incentives that are aligned with broader public policy goals, such as reducing congestion or traffic accidents.

Finally, with regards to the institutional drivers, the main determinants of MaaS are the following: Data privacy, public versus private, and regulation.

With the advent of digital platforms and digital solutions for the mobility sector, there has been a growing concern over data privacy. Rather than just being a driver, data privacy can be a barrier to the more widespread adoption of digital solutions, and, particularly, to the integration of solutions. Furthermore, the Public vs. Private initiative means that in most cities nowadays, private and public initiatives coexist together, making it more difficult to integrate solutions with distinct business models, objectives, and operating models. The more disruptive solutions, together with those that have the greatest impact, have been privately led, which raises concerns with regards to regulation. Consequently, the regulation of digital platforms and data collection, as well as usage and ownership, is a topic that has been attracting increasing attention. However, the regulatory issues are far more complex, with an additional complexity being how to accommodate disruptive business models and innovative mobility solutions nowadays within existing rigid regulatory models. For example, traditional public transport companies are based on prescriptive concession contracts, whose re-negotiation or flexibility can be extremely difficult (for more on re-negotiation, see [39,40]).

## 4. MaaS Solutions

Over the last few years a growing number of MaaS solutions have emerged, which offer distinct services and options for users. These solutions have been primarily concentrated in Europe and the US, although there is a lower level of development in the latter. Table 2 presents an overview of several MaaS solutions from the following countries: France, the USA, the UK, Canada, Australia, Spain, Sweden, Germany, and the Netherlands.

One example of a MaaS solution is Ustra, a mobility shop which provides a channel for the sale of physical tickets for transport operators. All services are based on a mobility app, which is essential for providing a pay-as-you-ride or a pay-as-you-go solution, which is often complemented by additional web-based services. The type of services offered is distinct, although these cover the following main features: (a) A transport planner—allows users to identify which transport solutions are available to connect the origin and destination of a journey, through a service which is generally provided by data aggregators, such as Google or Apple, although many smaller, locally-based solutions also exist which offer transport planning solutions with real time information on the system; (b) booking services—allows users to book tickets for regional/long distance trains in advance; (c) parking—whereby users can park with payment being made through a MaaS app; (d) bike-sharing—where certain MaaS solutions integrate bike-sharing service, whereby the utilisation of the bike at the docking station is activated by the app; (e) car rental—provides the ability to book and pay for rental cars, and; (f) taxis—provides the ability to book (if necessary and when available) and pay for a taxi ride.

MaaS is not a binary concept, and accordingly the reality is that there are many variances and different layers for the adoption of these solution. Table 3 presents the existing levels of MaaS. Nowadays, the most common levels found in cities are levels 0, 1, and 2, with Levels 5 and 6 still being part of the medium to long term vision of most cities. Certain experiences are understood to be Level 3, however, a single account does not usually fully integrate the various available mobility services, particularly those that operate on a stand-alone basis—such as shared services.

Most solutions are still in levels 2 and 3, such as Optymod, Whim, Ubigo, Moovel, or Mobility Mixm working as mobility integrators where instead of having multiple channels, the interface is unified across the modes, provider, and services that the traveller finds necessary for journeys.

Ustra does provide a higher level of integation, providing a mobility shop, and requiring minimal intervention from the user. Levels 5 and 6 are still not provided on a commercial basis.

**Table 2.** Examples of MaaS solutions.

| Name | Optymod | TransitApp | Whim | Mobility 2.0 Services | Ubigo | Ustra | Moovel | Mobility Mix |
|---|---|---|---|---|---|---|---|---|
| *City/Country* | Lyon/France | USA, UK, Canada, Europe and Australia | Helsinki/Finland | Palma/Spain | Gothenburg/Sweden | Hannover/Germany | Berlin/Germany | Netherlands |
| *Beginning of operation* | 2012 | 2012 | 2016 | 2013 | 2016 | 2016 | 2012 | 2000 |
| *Main function* | Mobility Integrator | Mobility Integrator | Mobility Integrator | Mobility Integrator | Mobility Integrator | Mobility Shop | Mobility Integrator | Mobility Integrator |
| *Product* | Mobile App | Mobile App + Website | Mobile App | Mobile App + Website | Mobile App | Mobile App + Physical Card Pass | Mobility App | Mobility App |
| *Service* | Transport planner, and booking for bike sharing, regional trains and parking | Pay-per-ride for public transport, bike and car sharing, taxi | Pay-per-ride for taxi drivers, car rental, bike sharing and public transport Mobility planner | Pay-per-ride for public transport, bike-sharing and taxi | Pay-per-ride for taxi drivers, car rental, bike sharing and public transport Mobility planner | Integrated mobility bill Pre-reserve taxi and car sharing | Pay-per-ride for mytaxi, public transport, car-sharing and bike-sharing. | Monthly travel budget Includes taxi, public transport, transport planning |

Source: Adapted from [24].

**Table 3.** Levels of MaaS.

| Level | Description | Explanation |
|---|---|---|
| 0 | Base level, corresponds to existing status quo in most cities. | There are account base systems, where individual models of transportation already have a digitalised interface and the traveller has information available online for each type of transportation. |
| 1 | There is one-to-one integration between some private services. | Duets of services which start to develop joint offering (e.g., tolls+car park, private car+ferry, and car + ride bus services). |
| 2 | Integrate payment and ticketing across modes of limited public and private modes of transportation services. | At this level, greater integration occurs, although this time between a private operator and a public transport mode of operation. Integration shows promise, but other PT modes are sceptical and continue to remain aloof. |
| 3 | Unified interface single account used in multiple modes of transport services. | Instead of having multiple channels, interface is unified across the modes, provider, and services that the traveller finds necessary for journeys, which are provided by a single meta-operator through a Traveller account. |
| 4 | All modes are integrated, private and public, including routing, ticketing, and payment. | Open data and standards are defined and commonly used by all transportation providers and MaaS meta-operators to provide services for Travellers. |
| 5 | Active artificial intelligent choices are taken based on travels preferences and near real time data for ad-hoc changes to a journey. | Based on traveller-specific behaviour and profiling, minimal (to none) intervention is needed by the traveller for an end-to-end journey—based on the traveller's preferences, past travel history, and filters. |
| 6 | MaaS connects beyond mobility, interfacing with internet of things (IoT), smart buildings, and smart cities. | As MaaS evolves, so do the other systems that are involved in the traveller's day, such as smart work spaces, smart homes, smart cities, and general services (e.g., food, groceries, entertainment, sport, culture) in order to provide convenient and seamless interface with the Traveller's eco-system. |

Source: Adapted from [25].

Reference [41] argues that the impact of MaaS strategies will not be limited to users' experience. Cities are going to suffer major changes at different levels. First, MaaS is expected to facilitate the integration and use of several distinct transport modes. This requires Governments to anticipate the need to invest in public transport to guarantee an increase in capacity and, consequently, ensure a smooth physical integration with other systems. It is important to consider that many public transportation systems have been under-financed and thus the issue of financing is increasingly important. Furthermore, with an increase in the use of shared services, the number of cars in circulation is likely to reduce, which will subsequently reduce revenues from car sales or fuel consumption, pressuring government budgets.

## 5. Case Study: Lisbon (Portugal)

This section provides a diagnosis and analysis of the use of MaaS-type apps in Lisbon, the capital of Portugal. The city of Lisbon has a population of 550,000 inhabitants, with 2.8 million residing in the Metropolitan Area. As in the case of most cities, inner urban mobility is provided by a metro and light rail systems, while the commuter transport is provided by several commuter trains and ferries, which articulate with several bus operators (public and private). Over the last five years, the system has evolved significantly due to the emergence of several new transport operators, both for car sharing, and also bicycles and scooters.

Over the last few years, each operator has been trying to improve their users' experience, as well as facilitate payments and integrate tariff systems. This pushed towards intermodal transport ticketing system, which is used by all the major public transport operators in Lisbon, although it does not integrate any other new transport services, neither does it permit a full digitalisation of the travel experience, as a physical travel card is still required. Indeed, the Viva System only enables ticket integration, whereas, on the other hand, the public-owned company that manages street parking in Lisbon (EMEL) has developed its own app for parking in any area of Lisbon, and it also provides bike-sharing technology (ePark).

On the private sector side, the most relevant initiative has been that of Via Verde, which was traditionally a digital payment system for paying motorway tolls, but has now expanded to other services, such as public transport (integrating a privately-operated commuter rail operator), parking, car-sharing, and it has even developed a consumer reward programme.

These apps also contain other services, which are essentially related with route planning and real time information that provide users with useful data for their daily travel. These are dominated by the traditional data aggregators, such as Google, Apple, Waze, etc. Figure 1 present the framework of the transport system of Lisbon, and Figure 2 lists the smart phone mobility apps that are required for its use and navigate within the transportation system in Central Lisbon.

The organic growth of mobility strategies and the absence of a dominant MaaS operator for the integration of all new services, has led to the emergence of several MaaS options. Transport operators are not wasting time and are developing their own apps. It is clear that in the medium and long term, mobility apps and MaaS strategies will merge into truly integrated services.

**Figure 1.** Transport operators in Lisbon.

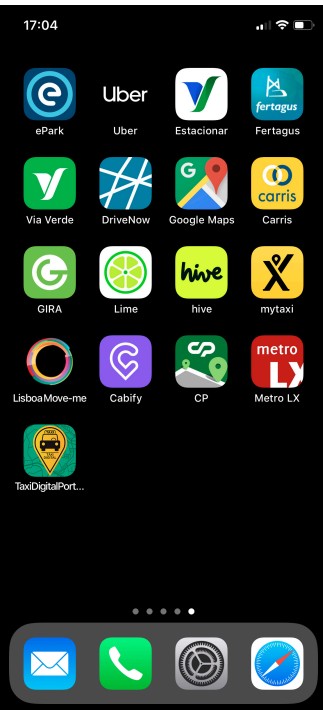

**Figure 2.** Apps on a Lisbon user's mobile phone.

To support the discussion of MaaS development in Lisbon, we have addopted a SWOT (Strength–Weakeness–Opportunities–Threats) analysis, highlighting the main strengths, weaknesses, opportunities, and threats, according to the well kown framework. The main dimensions of the analysis are presented bellow:

- Strengths: The development of several apps has allowed a high level of maturity in existing solutions; the apps are well established, and users are becoming familiar with the digitalization of the transport experience; there is growth in public transport use (at least up to the COVID pandemic);
- weaknesses: Several solutions that are fragmenting the market and creating an additional complexity for passengers;
- opportunities: Mobility patterns less stable and more fuzzier, thus requiring more complex and tailor-made solutions; familiarity with digital payment solutions is increasing; there is an overall trend digitalization of the economy; decentralization of management (from central to local government), which has allowed for a faster and more effective decision making process; emergence of non-traditional mobility solutions, based on the sharing economy; the need for decreasing physical payments as a consequence of COVID-19, and also the additional flexibility in terms of mobility that has emerged from the pandemics;
- threats: Reluctance of players to have control over their own apps; conflicting objectives between private and public owned companies; unclear regulatory framework and data privacy issues.

However, several questions remain, particularly regarding the nature and the number of MaaS operators, and also the regulation of MaaS services. As illustrated above in the SWOT analysis, the strengths and opportunities clearly outnumber the weaknesses and threats, but the latter are creating significant barriers. For example, regarding the fragmentation of the market, it is unclear whether the dominant MaaS operator will be one of the large data aggregators (i.e., Google or Apple), or one of the emerging smaller mobility integrators in each city. However, at the moment, and considering all the uncertainties involved, it is very difficult to forecast whether the solution will be a mix of both types of companies, organised in city-based partnerships. There will be very few truly global MaaS providers in

the next 10 to 15 years, as each system is subject to several specificities, an example being the ticketing system and physical control/entry points. For instance, the adaptation of existing physical overhead gantries for the use of mobile apps involves significant investment by public transport operators, which are facing chronical underfinancing.

However, the regulation of MaaS Services is critical, as these systems will be the pivotal stakeholder for the management and planning of urban mobility systems. The real challenge is how to control data privacy and data usage, and, also how to ensure that this data is readily available for monitoring the performance of the system and each operator, and for contributing to the provision of public policy strategies which are capable of maximising welfare overall.

## 6. Conclusions

It will still take some time to fully implement the true concept of MaaS at its most advanced level. For the existing status quo is still based on a competitive strategy, rather than a pooling of MaaS operators, and it is still unclear who will be the winners. It is also unlikely that any of the existing operators will become the dominant operator under their existing business models. Indeed, some form of partnerships between global data aggregators and mobility integrators will be required if Maas is to be truly integrated. Nevertheless, the urban mobility ecosystem is evolving and is testing alternative strategies, albeit with different strategies, across different geographic regions.

In Europe, this process has been managed by central governments, which recognise the potential of MaaS for contributing to reducing car trips and for enhancing public transport by making it more "seamless" and "painless", as argued by [25]. Many barriers still need to be overcome, such as the issues of interoperability and shared information with governments [42].

In the US, the positioning is different. MaaS has been primarily driven by local governments, and is more focussed on car sharing services, including autonomous vehicles. There is no objective to reduce car usage or to improve public transport efficiency, but rather the focus is on providing new business models and an improved utilisation of cars.

MaaS also requires new mobility behaviours from users. As [5] discusses, there are specific reasons why people resist change regarding their mobility behaviour, examples being a lack of awareness of the total costs when comparing mobility alternatives, or a lack of information on public transport and shared transport alternatives, social ties, and personal preferences. This simultaneously represents a barrier to the adoption of MaaS, and yet it is one of the main reasons why MaaS strategies are so badly needed. Nevertheless, as the SWOT analysis for the Lisbon case illustrates, the existing strengths and opportunities outweigh the weakenesses and threats, although data privacy and regulatory issues are a significant to be dealt. Dematerialization and digitalization of the mobility experience is a need, as well as system that allows a closer monitoring of passengers' movements. The recent COVID-19 crisis has provided additional pressure on this pathway.

**Author Contributions:** Conceptualization, C.O.C. and J.M.S.; methodology, C.O.C. and J.M.S.; formal analysis, C.O.C. and J.M.S.; writing—original draft preparation, C.O.C. and J.M.S.; writing—review and editing, C.O.C. and J.M.S.; All authors have read and agreed to the published version of the manuscript.

**Funding:** We are gratefully acknowledge financial support from FCT—Fundação para a Ciencia e Tecnologia (Portugal), national funding through research grant UIDB/04521/2020.

**Conflicts of Interest:** The authors declare no conflict of interest.

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
