# Peer review of "“Mobility as a Service” Platforms: A Critical Path towards Increasing the Sustainability of Transportation Systems"

_sustainability, doi:10.3390/su12166368_

Round 1
Reviewer 1 Report
The article is a conceptual paper that deals with an issue of absolute importance: the dystonia between the development of the Maas systems from the point of view of technology and engineering of transport systems and their actual application. The authors highlight the evidence of limits and barriers that, according to their typology (technological, legal, social, market-based...) constitute a concrete slowdown in the affirmation of the scientific paradigms that represent the engine of the Maas revolution. A complete and up-to-date state of the art is provided and the methodological approach is clear, although not directly explained by the authors.
The article is well written, even if there are, in my opinion, some shortcomings, which I am going to list below.
- The abstract does not highlight the objective of the work, merely outlining some of the background lines of the Maas systems and their main opportunities and weaknesses. It would be appropriate, in my opinion, to highlight at an early stage the elements that the authors intend to address and the objectives of the article.
- Paragraph 5, Lisbon case study, needs to be expanded and improved. The authors, having reviewed the Maas systems and their characteristics in great detail in the previous paragraphs, intend to highlight in this paragraph the presence of elements that hinder the concrete application of these integrated transport solutions. The paragraph describes the different Maas services and operators present in the city of Lisbon and the authors underline, as critical points, precisely their high number and the absence of a dominant operator that can act as an aggregator of services. In addition, other limitations are those linked to the cost of developing apps and the usual well-known privacy issues. All these, and others, could be included in some qualitative analysis of strategic planning (from a basic SWOT to a PEST or PESTEL). This would make the arguments more solid and allow readers to focus more easily on the main elements that limit the practical development of the Maas systems and their mutual interactions.
- In the light of any changes to paragraph 5, the conclusions should be adapted and expanded.
Some minor observations and typos:
- check the font size, because in many parts of the text they seem to be different (e.g.: page 1, lines 33-36);
- Page 5, line 172 "forthe"
- Page 7, 232-240 I would reverse the priorities expressed in the sentence;
- Page 8. Line 286 missed reference tab2
- Page 10 line 308 missed reference tab 3
- Page 12 lines 360-61 missed references fig 1 and 2
Reviewer 2 Report
In this work, the authors present an analysis of the determining factors of MaaS, its barriers and existing solutions.
The work is well written in general, although some aspects of English grammar can be improved and revised (for example, the last paragraph of section 4)
Regarding the first objectives (drivers and barriers), the results are adequate. However, there is an extensive biography on MaaS with many studies on different technological, economic and social elements (same references in this journal), and I am not very clear about the novel contributions of the manuscript (you can specify). I miss two aspects of the analysis carried out. The first, not finding Data Analytics or Big Data among the fundamental technological factors that enable change. The second that does not delve into sustainability, being the central topic of the magazine. For example, the importance of MaaS not included in the reduction of the private vehicle and the modal change to public transport. MaaS is said to have the potential to internalize the externalities of several transport modes on a global scale. But does not specify what those externalities are.
On the other hand, the objective of analyzing the existing systems remains quite weak. I cannot find a detailed analysis. Only you explain an example, and the other solutions are not analyzed. Also, in the last paragraph, I do not know very well its relationship with the solutions examined. In the case of study, something similar also happens, the last two paragraphs seem more conclusions and general arguments (some already explained above) than of the case of Lisbon.
Minor changes:
You must define all the initials to facilitate reading for all kinds of readers. On line 212, PPP (Public-Private Partners) is not.
There are several references in the text with errors: line 286, 308, 360, 361
Authors must review bibliography. There are some typos: lines 433, 439, 461, 476, and a repeated reference (lines 467,470)
Reviewer 3 Report
Thank you for the opportunity to review this interesting paper on MaaS. Overall this is a wonderful paper. However, I do have some comments below:
The authors are discouraged from using the term "ridehailing" as this is inconsistent with international terms and definitions (i.e., transportation network companies/TNCs which have been translated into numerous languages such as Voitures de Transport avec Chauffeur (VTCs). Please also use consistent spelling for all terms/definitions (e.g., one word vs. two words/hyphens, etc.).
Consider replacing "solutions" with "strategies" - Solutions has an underlying advocacy tone.
If shared mobility is one of the "backbones" of MaaS, is there a reason why shared mobility literature is not referenced? A couple USDOT reports that could be helpful:
There are a few reference errors that should be corrected in the manuscript.
There is a large body of literature about on-demand mobility, MaaS, and the commoditization of travel that is not referenced and should be. Some helpful resources could include:
https://www.sciencedirect.com/science/article/pii/B9780128150184000036
https://rosap.ntl.bts.gov/view/dot/34258
https://www.nxtbook.com/ygsreprints/ITE/ITE_June2020/index.php?startid=29#/p/28
Round 2
Reviewer 1 Report
After the first round of review the article has been improved and the comments highlighted have been accepted. In particular, the addition of some analysis - albeit qualitative - to the Lisbon case study makes the work more rigorous and above all more useful for potential readers. In particular, the SWOT analysis directs, at a preliminary level, towards concrete lines of approach for the solution of the limits highlighted to the implementation of MaaS solutions.
Even the abstract is now more focused on the specific content of the analysis. The article remains conceptual and very general, but it is well written and has the merit of highlighting the potential and real limits to the implementation of MaaS solutions. A significant limitation is the limitation of the case study to the city of Lisbon alone. However, the work is of good quality and, in my opinion, worthy of publication. Probably the last sentence on the pressures generated by the COVID-19 outbreak added to the conclusions, while correct, would perhaps deserve some additional consideration (in the SWOT analysis part), although the issue is of enormous importance and I recognise that it would certainly require more space and dedicated articles.
I recommend a final text check.
Reviewer 2 Report
The work has improved, most of the proposed aspects have been corrected. Although the authors could make a little more effort around the comments made.
When rereading the text, the authors' references seem excessive, with all the existing references (some from the journal and editorial) such as:
Nikitas, A.; Kougias, I.; Alyavina, E.; Njoya Tchouamou, E. How Can Autonomous and Connected Vehicles, Electromobility, BRT, Hyperloop, Shared Use Mobility and Mobility-As-A-Service Shape Transport Futures for the Context of Smart Cities? Urban Sci. 2017, 1, 36.
Nikitas, A.; Michalakopoulou, K.; Njoya, E.T.; Karampatzakis, D. Artificial Intelligence, Transport and the Smart City: Definitions and Dimensions of a New Mobility Era. Sustainability 2020, 12, 2789.
Santos, G. Sustainability and Shared Mobility Models. Sustainability 2018, 10, 3194.
Keller, A.; Aguilar, A.; Hanss, D. Car Sharers’ Interest in Integrated Multimodal Mobility Platforms: A Diffusion of Innovations Perspective. Sustainability 2018, 10, 4689.
Harrison, G.; Gühnemann, A.; Shepherd, S. The Business Case for a Journey Planning and Ticketing App—Comparison between a Simulation Analysis and Real-World Data. Sustainability 2020, 12, 4005.
Matyas, M., Kamargianni, M. The potential of mobility as a service bundles as a mobility management tool. Transportation 46, 1951–1968 (2019). https://doi.org/10.1007/s11116-018-9913-4
Some essential changes:
English grammar not only the last paragraph of section 4
References in the bibliography have not been corrected: (Now in lines 535 and 541) that there are excess scripts separating “Bri-to” and “Euro-pean and a repeated reference (now in lines 571 and 573)
